# MonoLift: Learning 3D Manipulation Policies from Monocular RGB via Distillation

**Ziru Wang**[1], **Mengmeng Wang**[*2,1], **Guang Dai**[1], **Yongliu Long**[3], **Jingdong Wang**[4]
[1]SGIT AI Lab, State Grid Corporation of China [2]Zhejiang University of Technology
[3]Zhejiang University [4]Baidu

## Abstract

Although learning 3D manipulation policies from monocular RGB images is lightweight and deployment-friendly, the lack of structural information often leads to inaccurate action estimation. While explicit 3D inputs can mitigate this issue, they typically require additional sensors and introduce data acquisition overhead. An intuitive alternative is to incorporate a pre-trained depth estimator; however, this often incurs substantial inference-time cost. To address this, we propose MonoLift, a tri-level knowledge distillation framework that transfers spatial, temporal, and action-level knowledge from a depth-guided teacher to a monocular RGB student. By jointly distilling geometry-aware features, temporal dynamics, and policy behaviors during training, MonoLift enables the student model to perform 3D-aware reasoning and precise control at deployment using only monocular RGB input. Extensive experiments on both simulated and real-world manipulation tasks show that MonoLift not only outperforms existing monocular approaches but even surpasses several methods that rely on explicit 3D input, offering a resource-efficient and effective solution for vision-based robotic control. The video demonstration is available on our project page: `https://robotasy.github.io/MonoLift/`.

## 1 Introduction

Robotic manipulation demands precise spatial understanding to support reliable 3D actions generation. While existing methods leverage multi-view images [1, 2], point clouds [3, 4], or RGB-D [5, 6] sensors to encode 3D structure, they typically require specialized hardware and complex preprocessing steps, including calibration, alignment, and filtering. These constraints limit scalability and hinder deployment in real-world, resource-constrained settings (see Figure 1(a)).

As a practical and lightweight alternative, recent efforts have increasingly focused on learning 3D manipulation policies directly from monocular RGB images [7, 8, 9, 10]. However, a fundamental gap remains between 2D visual perception and 3D action reasoning: ***visual similarity in 2D does not imply consistency in 3D actions***. In manipulation tasks, visually similar observations may correspond to states that demand distinct actions, which often stem from subtle positional or structural variations. Without access to such structural cues, imitation learning models frequently fail to disambiguate these cases, leading to ambiguous or suboptimal actions and ultimately hindering the learning of precise state–action mappings [11, 12, 13]. Recent methods attempt to address this issue by enriching monocular representations with implicit structural cues (see Figure 1(b)). These efforts span a range of strategies, including modeling temporal dynamics through video or image prediction [7, 9] and synthesizing novel views from a single frame to inject spatial priors [10]. While these methods have shown promise, they often adopt multi-stage pipelines, where structural reasoning is performed as an

---

[*]Corresponding Author

intermediate step separate from policy learning. Such indirect designs often introduce cumulative errors and weaken the coupling between perception and decision-making.

A straightforward solution is to integrate depth estimators into the policy architecture, using their outputs as geometry-aware inputs. With recent advances in monocular depth estimation [14, 15, 16, 17], it is now feasible to obtain pseudo-depth maps from RGB images. However, incorporating such estimators into the inference pipeline significantly increases computational cost and latency (see Figure 1(c)), limiting their practicality in real-world deployment. This raises an important question: *Can we retain the benefits of depth-guided 3D reasoning without incurring inference-time cost?*

We answer this with **MonoLift**, which lifts monocular RGB inputs into 3D-aware perception and control by distilling knowledge from a depth-guided teacher (see Figure 1(d)). To enable effective distillation, we identify three fundamental challenges in policy learning from RGB-only inputs: (i) difficulty in spatial disambiguation, (ii) limited temporal cues, and (iii) misguided actions due to absent 3D priors.

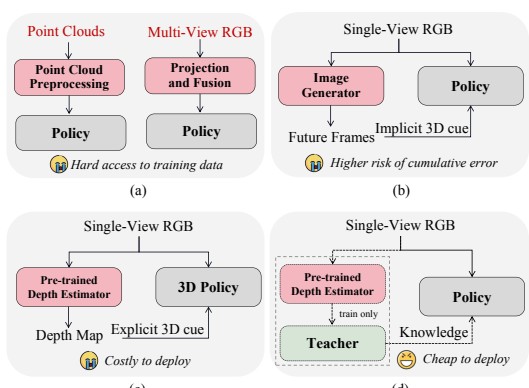

Figure 1: **Comparison of frameworks for 3D policy learning.** Our method (d) leverages 3D cues estimated by off-the-shelf depth estimators during training via distillation, requiring no external 3D data (as in a) and no inference-time modules (as in b and c). Red indicates additional inputs or modules beyond single-view RGB.

These observations motivate our tri-level knowledge distillation framework, which targets each limitations through three complementary components: *(i) Spatial Representation Distillation:* Transfers fused RGB–depth features from the teacher to help the student disambiguate visually similar yet structurally different observations. *(ii) Temporal Dynamics Distillation:* Aligns temporal feature trajectories to enable the student to capture motion patterns that reflect underlying 3D structural changes. *(iii) Action Distribution Distillation:* Transfers action distributions shaped by the teacher's 3D understanding, guiding the student to generate geometry-aware behaviors. *Our ablation study in Figure 5(a) confirms each distillation level provides distinct gains.*

In summary, the contributions of this work are threefold: (i) We propose MonoLift, a resource-efficient policy learning framework that learns from a 3D-aware teacher built on a pre-trained depth estimator, enabling monocular RGB agents to perform structured perception and control without extra 3D data or inference overhead. (ii) We design a tri-level knowledge distillation strategy that conveys spatial, temporal, and behavioral cues to improve the student's contextual understanding and decision-making under limited-modality constraints. (iii) We validate MonoLift across a wide range of simulated and real-world robotic manipulation tasks, demonstrating its ability to effectively learn 3D-aware policies while maintaining deployment efficiency.

## 2 Related Work

**Leveraging Explicit 3D Inputs for Policy Learning.** To improve spatial reasoning and control precision in complex environments, many works incorporate explicit 3D inputs such as RGB-D images, multi-view observations, and point clouds. Some methods focus on constructing coherent 3D representations to enhance spatial understanding [18, 19, 20, 21, 5]; for example, MV-MWM [1] and SPA [2] utilize multi-view masked autoencoders to learn geometry-aware embeddings. In addition, several methods integrate 3D perception directly into policy generation [22, 23, 24, 25, 26]. For instance, 3D Diffusion Actor [27] formulates trajectory generation as a diffusion process in 3D space, while 3D-VLA [28] unifies vision, language, and action modalities to support end-to-end planning. Despite their strong performance, these methods typically require high-cost sensors and complex processes, which limit their scalability. In contrast, our work targets a monocular RGB-only setting, aiming to learn 3D manipulation policies directly from single-view RGB images.

**Inferring Implicit 3D Cues for Policy Learning.** Policy learning for 3D manipulation under monocular RGB input has received increasing attention. Many methods directly map RGB inputs

to actions [11, 12, 13]. While efficient, they struggle to disambiguate visually similar states with different 3D structures, limiting control performance. To mitigate this, recent work has focused on inferring implicit 3D cues from monocular inputs. One common approach leverages temporal modeling by generating video sequences or intermediate frames to capture inter-frame disparities [29, 8, 30]. For example, AVDC [7] extracts actions from generated videos using rigid transformations computed from optical flow, while GROUND [9] aligns video models to continuous action spaces via goal-conditioned exploration. Other works synthesize pseudo-novel views from single RGB images using pre-trained diffusion models to inject explicit spatial priors [10]. Additionally, vision foundation models have been used to extract semantic and structural representations from static images [31, 32, 33], as demonstrated in MT-R3M [34], which combines R3M embeddings with a Transformer-based policy head. Although these methods show promise, they typically rely on multi-stage pipelines that introduce error accumulation and weak perception-action coupling. In contrast, our method leverages depth estimators to provide explicit geometric priors and adopts a multi-level distillation for efficient policy learning.

## 3 Problem Formulation

We aim to learn a policy from monocular RGB images, without relying on 3D sensors. Each training demonstration consists of a language instruction and an expert trajectory: $\tau_i = \left(g^i, \{(o_t^i, a_t^i)\}_{t=1}^T\right)$, where $g^i$ denotes the language command associated with trajectory $i$, $o_t^i$ is the RGB observation at time step $t$, and $a_t^i$ is the corresponding expert action. The full dataset is denoted as $\mathcal{D} = \{\tau_i\}_{i=1}^N$. The conditional policy $\pi_\theta(a_t \mid o_t, g)$ is modeled as a distribution over actions given the current observation $o_t$ and instruction $g$, enabling uncertainty-aware imitation [12, 13]. It is optimized via maximum likelihood estimation over expert demonstrations:

$$\mathcal{L}_{\text{actor}} = -\mathbb{E}_{(o_t, a_t^*, g) \sim \mathcal{D}} \left[\log \pi_\theta(a_t^* \mid o_t, g)\right], \tag{1}$$

where $\theta$ denotes the learnable parameters and $a_t^*$ is the ground-truth expert action. To enhance 3D awareness, we incorporate a teacher–student distillation framework as an additional training objective. Complementing the actor loss $\mathcal{L}_{\text{actor}}$, which captures expert behaviors, the proposed distillation losses (§ 4.2) convey spatial, temporal, and action-level cues from a depth-guided teacher.

## 4 Method

The overall architecture and data processing pipeline of MonoLift are presented in Section 4.1, while the details of the tri-level knowledge distillation mechanism are elaborated in Section 4.2.

### 4.1 Data Flow and Model Architecture

MonoLift consists of two main components: a **student model** for deployment and a **teacher model** for enhanced policy training (see Fig. 2).

**Student Model.** The student model processes a sequence of $H$ RGB frames (representing the visual history) and a language instruction through a pipeline composed of the following components:

- *Spatial Encoding*: Encode the RGB history using a ResNet-18 encoder to obtain $F_{\text{spa}}^{\text{S}}$. Encode the instruction into a language token using a 6-layer version of MiniLM provided in [35].
- *Temporal Modeling*: Concatenate visual features with the language token and append learnable action tokens to form the input sequence. This sequence is fed into a causal Transformer decoder, producing temporally-aware features $F_{\text{tem}}^{\text{S}}$ from observation tokens.
- *Action Prediction*: Use the Transformer outputs corresponding to action tokens as input to a MLP policy head, which generates the student's action distributions $A_{\text{act}}^{\text{S}}$.

**Teacher Model.** Used only during training, the teacher enhances visual representations by generating pseudo-depth signals. Its processing consists of:

- *Spatial Encoding*: For each RGB, generate pseudo-depth maps using a pretrained depth estimator (Depth Anything V2 [14]). Fuse the RGB–depth pairs using a unified encoder with a dual-path cross-modal fusion module (§4.2.1) to produce features $F_{\text{spa}}^{\text{T}}$ for spatial representation distillation.

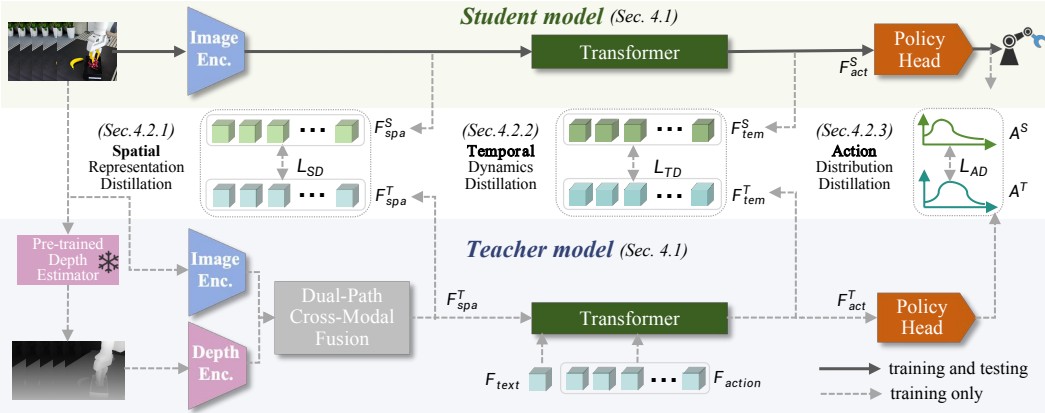

Figure 2: **Overview of the proposed MonoLift framework.** The student uses single-view RGB input, while the teacher incorporates estimated depth to guide spatial, temporal, and action learning. *Both models are trained end-to-end with shared encoder, Transformer, and policy head for consistent knowledge transfer.* To avoid redundancy, action and language tokens, shared between teacher and student, are depicted only in the teacher.

- *Temporal Modeling*: Concatenate each fused visual feature with the language token, and append learnable action tokens to form the input sequence. Pass this sequence through the Transformer decoder to model temporal evolution, yielding $F_{\text{tem}}^{\text{T}}$ for guiding temporal dynamics distillation.

- *Action Prediction*: Pass the action token outputs through the policy head to produce the teacher action $A_{\text{act}}^{\text{T}}$, which serve as behavioral guidance for the student.

## 4.2 Tri-level Knowledge Distillation Mechanism

Models based on monocular RGB inputs often lack structural awareness, limiting their effectiveness in manipulation tasks. First, without explicit geometric cues, the model struggles to infer spatial layouts and object relationships, impairing scene understanding. Second, since state transitions are often accompanied by spatial changes, the lack of structural cues hinders accurate temporal modeling and transition recognition. Third, lacking spatial grounding, the model tends to produce conservative or suboptimal actions, particularly in cluttered or visually ambiguous environments.

To tackle these limitations, we introduce a tri-level knowledge distillation strategy that transfers spatial, temporal, and action-level knowledge, enabling the student to acquire the teacher's 3D-aware capabilities: *(i) Spatial Representation Distillation* (§4.2.1) minimizes the discrepancy between the student's spatial features $F_{\text{spa}}^{\text{S}}$ and the teacher's depth-enhanced representations $F_{\text{spa}}^{\text{T}}$. *(ii) Temporal Dynamics Distillation* (§4.2.2) aligns the temporal gradients between teacher and student features $(\nabla_t F_{\text{tem}}^{\text{T}}, \nabla_t F_{\text{tem}}^{\text{S}})$ to enhance transition-aware representation learning; *(iii) Action Distribution Distillation* (§4.2.3) minimizes the KL divergence between the student's policy $A_{\text{act}}^{\text{S}}$ and the teacher's output $A_{\text{act}}^{\text{T}}$, transferring action distributions to encourage geometry-aware behaviors.

### 4.2.1 Spatial Representation Distillation

Monocular RGB lacks key geometric cues, such as depth boundaries and spatial layouts, making it challenging to infer object positions and occlusions. To address this, we adopt spatial representation distillation, where the teacher encodes both RGB and pseudo-depth using a *Unified RGB-Depth Encoder* with a *Dual-Path Cross-Modal Fusion* module. This process produces fused features $F_{\text{spa}}^{\text{T}}$, which help the student learn geometry-aware representations from RGB alone.

**Unified RGB-Depth Encoder.** To reduce low-level discrepancies between RGB images and pseudo-depth maps, we first render each depth map $d_t$ into a pseudo-colored heatmap to visually resemble its corresponding RGB frame $o_t$. This preprocessing step enables appearance-level alignment across modalities. Both RGB and pseudo-depth inputs are then passed through the shared image encoder $\mathcal{E}(\cdot)$ with identical weights. Despite parameter sharing, the encoder preserves modality-specific cues through input-driven activation differences, yielding spatial features $F_{\text{rgb}, t}^{\text{T}}$ and $F_{\text{dep}, t}^{\text{T}}$.

**Dual-Path Cross-Modal Fusion.** Using the features $F_{\text{rgb},t}^{\text{T}}$ and $F_{\text{dep},t}^{\text{T}}$, we employ a dual-path fusion strategy that combines attention-based and pointwise interactions to balance global integration and local structural consistency. Specifically, we concatenate $F_{\text{rgb},t}^{\text{T}}$ and $F_{\text{dep},t}^{\text{T}}$ and feed them into a multi-head self-attention module to produce $F_{\text{attn},t}^{\text{T}}$. We then split the attention output into two modality-specific halves and compute their per-token average to obtain $F_{\text{attn-mean},t}^{\text{T}}$, encouraging balanced contributions from both modalities. Simultaneously, we compute the token-wise average of RGB and depth features to obtain $F_{\text{avg},t}^{\text{T}}$, which maintains pixel-level alignment. The final fused representation is computed as: $F_{\text{spa},t}^{\text{T}} = \frac{1}{2}\left(F_{\text{attn-mean},t}^{\text{T}} + F_{\text{avg},t}^{\text{T}}\right)$. This fusion scheme retains modality alignment while enhancing geometry-sensitive features, resulting in enhanced spatial representations.

**Distillation Objective.** The student receives only RGB input and extracts spatial features $F_{\text{spa},t}^{\text{S}}$ using $\mathcal{E}(\cdot)$. To transfer the spatial knowledge encoded in the teacher's fused features $F_{\text{spa},t}^{\text{T}}$, we define a distillation objective that minimizes the discrepancy between the student and the teacher's output:

$$\mathcal{L}_{\text{SD}} = \frac{1}{H} \sum_{t=t'-H+1}^{t'} \left\| F_{\text{spa},t}^{\text{S}} - F_{\text{spa},t}^{\text{T}} \right\|_2^2, \tag{2}$$

where $H$ denotes the length of the historical window, and $t'$ represents the current prediction step. The loss is computed by averaging the feature alignment errors over the $H$ time steps.

### 4.2.2 Temporal Dynamics Distillation

Manipulation tasks often appear visually continuous, but temporal variations in RGB are insufficient to capture distinct state transitions. To address this, we introduce a temporal dynamics distillation mechanism that uses a *temporal modeling* module to capture state evolution over time and aligns *temporal gradients* between the teacher and student. This alignment enables the student to better capture structural changes and transition dynamics.

**Temporal Modeling Module.** We use a Transformer decoder as the temporal backbone for both teacher and student models. It consists of $L$ stacked blocks with masked self-attention, cross-attention, and feedforward layers, following the standard decoder design [36, 37, 38]. Sharing the decoder ensures consistent temporal encoding and efficient feature alignment without extra parameters.

At each decision step $t$, the agent processes a history of length $H$. For each timestep in this history, the decoder input comprises three types of tokens concatenated sequentially: (i) *Spatial tokens* $[F_{\text{spa},t-H+1}^{(\cdot)}, \ldots, F_{\text{spa},t}^{(\cdot)}]$, extracted by fusing RGB and depth inputs (teacher) or from RGB-only inputs (student); (ii) *Action tokens* $[F_{\text{action},t-H+1}, \ldots, F_{\text{action},t}]$, which are learnable embeddings initialized randomly and trained jointly to capture autoregressive action dependencies; (iii) *Language token* $F_{\text{text}}$, derived from the task instruction.

The decoder first applies masked self-attention to model temporal dependencies within this token sequence, followed by cross-attention to integrate contextualized observations from encoded perceptual features. The decoder then outputs a sequence of contextualized temporal features $[F_{\text{tem},t-H+1}^{(\cdot)}, \ldots, F_{\text{tem},t}^{(\cdot)}]$ to capture the dynamic evolution of visual observations, which serve as the primary representations for temporal dynamic distillation. Additionally, the decoder outputs contextualized action features $F_{\text{act},t}^{(\cdot)}$, which are passed to the policy head for action prediction.

**Distillation Objective.** Inspired by temporal gradient modeling in video understanding [39], we treat the variation of observation features over time as a key indicator of dynamic transitions in robotic manipulation. To transfer this capability, we align the temporal gradients of contextualized features output by the Transformer decoder, enabling the student to model the temporal evolution patterns encoded in the teacher's 3D-aware representations. The temporal gradients of the contextualized features from the teacher and student models at timestep $t$ are defined as $G_t^{\text{T}} = \nabla_t F_{\text{tem}}^{\text{T}}$, $G_t^{\text{S}} = \nabla_t F_{\text{tem}}^{\text{S}}$. Since the feature sequence is defined over discrete timesteps, the gradients are approximated using central differences. The temporal distillation loss is computed over the central $H - 2$ steps of the temporal window ending at the step $t'$:

$$\mathcal{L}_{\text{TD}} = \frac{1}{H-2} \sum_{t=t'-H+2}^{t'-1} \left\| G_t^{\text{S}} - G_t^{\text{T}} \right\|_2^2, \tag{3}$$

Table 1: **Quantitative results on 8 tasks from two Libero scenes.** Input types: **S-RGB** = Single-view RGB, **M-RGB** = Multi-view RGB, **RGB-D** = RGB+Depth. Task names are abbreviated: R/W = Red/White mug, Y/W = Yellow/White mug, C = Chocolate, L = Left, R = Right, P = Plate.

| Method | Input Type | R-L | R-R | W-L | Y/W-R | C-L | C-R | R-P | W-P | Avg. S.R. (%) |
|--------|-----------|-----|-----|-----|-------|-----|-----|-----|-----|---------------|
| MT-ACT | S-RGB | 43.3 | 20.0 | 70.0 | 36.7 | 20.0 | 20.0 | 33.3 | 40.0 | 35.4±2.6 |
| RT-1 | S-RGB | 33.3 | 36.7 | 73.3 | 70.0 | 30.0 | 40.0 | 46.7 | 50.0 | 47.5±2.7 |
| MT-R3M | S-RGB | 36.7 | 46.7 | 73.3 | 60.0 | 50.0 | 50.0 | 36.7 | 40.0 | 49.2±1.6 |
| GROUND | S-RGB | 38.4 | 40.8 | 51.2 | 38.4 | 70.4 | 79.2 | 72.8 | 25.6 | 52.5±8.8 |
| 3D-VLA | RGB-D | 56.7 | 46.7 | 73.3 | 66.7 | 86.7 | 83.3 | 60.0 | 76.7 | 68.7±1.0 |
| SPA | M-RGB | 40.0 | 36.7 | 70.0 | 76.7 | 70.0 | 76.7 | 60.0 | 60.0 | 61.2±2.7 |
| **MonoLift** | S-RGB | 63.3 | 83.3 | 83.3 | 73.3 | 80.0 | 100.0 | 83.3 | 80.0 | **80.8±3.3** |

where the boundary steps are excluded to avoid the influence of edge effects, ensuring stable gradient computations over the temporal window.

### 4.2.3 Action Distribution Distillation

In standard imitation learning, expert labels provide deterministic action targets but fail to capture potential uncertainties and correlations among different actions. In contrast, a teacher equipped with RGB-D inputs can generate a more informative action distribution that reflects a deeper spatial understanding and richer 3D structural information, producing more stable and consistent action sequences. Therefore, we introduce action distribution distillation, which minimizes the Kullback–Leibler (KL) divergence between the teacher and student action distributions, enabling the student model to learn 3D-aware decision-making even when only RGB inputs are available.

**Policy Head Module.** Following temporal modeling, we employ a multi-layer perceptron (MLP) with $M$ linear layers as the policy head shared by both teacher and student models. This module takes temporally contextualized action features $F_{\mathrm{act},\,t}^{(\cdot)}$ as input and outputs parameters defining a Gaussian action distribution $A_t^{(\cdot)}$, represented by its mean and standard deviation.

**Distillation Objective.** To align action distributions, we minimize the KL divergence at each step $t$, conditioned on a window of $H$ past observations:

$$\mathcal{L}_{\mathrm{AD}} = \mathrm{KL}\left(A_t^{\mathrm{T}} \,\|\, A_t^{\mathrm{S}}\right), \tag{4}$$

where $A_t^{\mathrm{T}}$ and $A_t^{\mathrm{S}}$ are the action distributions produced by the teacher and student, respectively.

MonoLift is trained end-to-end using a total loss function that combines the three distillation objectives, along with the actor loss as defined in Equation 1.

## 5 Experiments

**Environments.** We evaluate our method on several simulated benchmarks and real-world robotic manipulation tasks, covering a diverse range of challenges: (i) LIBERO-90 [40] for visually ambiguous tasks involving subtle structural differences; (ii) Meta-World [41] for fine-grained manipulation; and (iii) LIBERO-LONG [40] for long-horizon tasks. See supplementary materials for details.

**Baselines.** We compare MonoLift against multiple baselines, as detailed in supplementary materials.

- **Single-view RGB methods (direct mapping):** *RT-1* [11] and *MT-ACT* [12] adopt direct imitation learning from single-view RGB observations, without incorporating any form of 3D information. They represent lightweight baselines suited for real-world deployment.
- **Single-view RGB methods (learned 3D cues):** *GROUND* [9] encourages implicit 3D understanding by predicting future visual observations. *MT-R3M* [34] incorporates spatial and structural information extracted by a pretrained visual encoder, R3M [31].
- **Explicit 3D input methods:** *3D-VLA* [28] and *SPA* [2] utilize explicit 3D inputs, such as depth maps or multi-view RGB images, to provide geometric priors for policy learning.

| Method | Input Type | Avg. S.R. (%) |
|---|---|---|
| MT-ACT | S-RGB | 11.3±0.5 |
| RT-1 | S-RGB | 67.6±1.6 |
| MT-R3M | S-RGB | 72.2±2.7 |
| GROUND | S-RGB | 74.8±1.6 |
| 3D-VLA | RGB-D | 83.2±2.4 |
| SPA | M-RGB | 72.8±2.1 |
| **MonoLift** | S-RGB | **87.8±2.3** |

(a)    (b)    (c)

Figure 3: **Quantitative results and qualitative analysis on Meta-World.** Success rates including (a) overall average and (b) individual results for easy, medium, and hard tasks. (c) Visualized failure cases of the RT-1, and corresponding successful executions in `bin_picking` by MonoLift, both under monocular RGB input.

## 5.1 Performance Comparision

**Evaluation on LIBERO-90.** We evaluate our method on the LIBERO-90 benchmark [40], where challenges arise for monocular RGB-based methods due to the difficulty in distinguishing visually similar objects in the absence of 3D information. For a fair comparison, we adopt the same 8 representative tasks from two diverse scenes in LIBERO-90, each paired with 20 expert demonstration trajectories, as outlined in [9]. We compare our method to three categories of baseline methods, introduced in the previous section under a unified experimental setup. As shown in Table 1, our approach demonstrates significant performance improvements across most tasks, achieving the highest average success rate among all evaluated methods. Compared to approaches that implicitly infer 3D cues from visual sequences, such as MT-R3M and GROUND, our method avoids multi-stage modeling and the cumulative errors associated with it by directly distilling from a depth-guided teacher. Moreover, it outperforms methods relying on ground-truth 3D inputs, such as 3D-VLA and SPA, which often lack explicit guidance in the spatial, temporal, and behavioral dimensions. Our joint distillation strategy combines these aspects into a tightly coupled perception-control framework that enhances both task understanding and execution in complex environments.

**Evaluation on Meta-World.** To assess the effectiveness of our method on tasks requiring fine-grained control, we evaluate it on 15 manipulation tasks from the Meta-World benchmark [41], covering diverse object interaction and tool-use scenarios. Based on control difficulty and geometric reasoning demands, we categorize the tasks into 7 easy, 5 medium, and 3 hard categories, with each task comprising 20 expert demonstrations (details provided in the supplementary materials). These tasks require complex 3D reasoning, which is particularly challenging under monocular RGB input, especially in our multi-task setup. As shown in Figure 3 (a-b), our method achieves consistently strong performance, with especially large gains on hard tasks, underscoring its robustness and precision in complex settings. Figure 3 (c) illustrates typical failure cases of RT-1 in the *bin_picking*. Due to the lack of 3D information, the model often struggles to localize objects accurately in scenarios with varying target positions, leading to misaligned grasping attempts. In contrast, our method successfully picks the target object under the same input conditions, demonstrating stronger spatial reasoning and robustness to target variations. Notably, this is achieved without any depth input during inference.

**Evaluation on LIBERO-Long.** To evaluate the effectiveness of our proposed MonoLift in long-horizon, multi-stage manipulation tasks, we conduct experiments

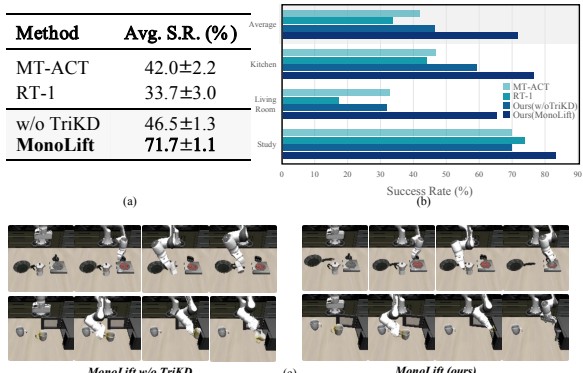

| Method | Avg. S.R. (%) |
|---|---|
| MT-ACT | 42.0±2.2 |
| RT-1 | 33.7±3.0 |
| w/o TriKD | 46.5±1.3 |
| **MonoLift** | **71.7±1.1** |

(a)    (b)

(c) MonoLift w/o TriKD   MonoLift (ours)

Figure 4: **Quantitative and qualitative results on the LIBERO-Long benchmark.** (a) Average success rates across all LIBERO-Long tasks under single-view RGB input. (b) Scene-wise comparison in three environments: *Kitchen*, *Living Room*, and *Study*. (c) Qualitative comparison of typical failures from the w/o TriKD variant, with successful completions by MonoLift under the same input.

on the LIBERO-Long benchmark [40]. Each task in

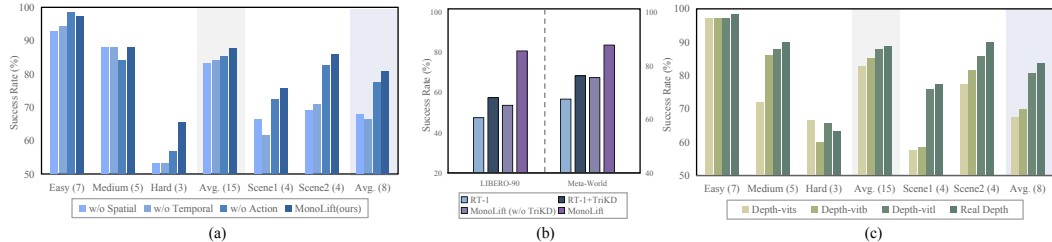

Figure 5: (a) Ablation Study on TriKD. (b) Effectiveness and Transferability of TriKD. (c) Impact of Depth Estimation Quality on Distillation Performance.

LIBERO-Long comprises a sequence of interdependent subgoals, posing significant challenges for spatial understanding and temporal stability. These experiments focus on evaluating policy performance under single-view RGB input. As shown in Figure 4 (a-b), our method consistently outperforms existing single-view RGB methods (MT-ACT, RT-1) as well as our ablated variant without distillation (w/o TriKD) across multiple scenarios. In such complex tasks, baseline methods often struggle to correctly transition between subtasks, resulting in unstable execution. Figure 4 (c) shows typical failure cases. Without 3D-aware distillation, the model fails to accurately localize the handle of the moka pot, resulting in the pot being knocked over. Additionally, it also fails to perceive the spatial structure of the microwave, causing the mug to be only partially placed and eventually dropped. In contrast, MonoLift demonstrates significantly better spatial understanding and policy stability under the same monocular RGB input.

## 5.2 Ablation Studies and Analysis

**Ablation Study on TriKD.** We ablate the tri-level knowledge distillation (TriKD) strategy in MonoLift by selectively disabling each guidance branch: spatial, temporal, and action-level. All variants share the same student architecture and training settings to ensure fair comparison. Specifically, we consider four model variants: the full TriKD model (ours) and three ablated versions without spatial (*w/o Spatial*), temporal (*w/o Temporal*), or action-level (*w/o Action*) distillation. As shown in Figure 5(a), the complete TriKD consistently achieves the best performance across all tasks, confirming the complementary nature of multi-level supervision. Removing spatial or temporal distillation leads to notable performance drops, particularly in complex scenarios. Spatial distillation guides the student in distinguishing geometrically distinct states; without it, the model often confuses similar-looking inputs and selects incorrect actions. Temporal distillation ensures consistent state transitions across time steps, while its removal results in unstable and incoherent behavior. Excluding action distribution distillation primarily affects the student's ability to learn the 3D-aware teacher's policy, leading to difficulties in selecting the correct actions.

**Effectiveness and Transferability of TriKD.** To assess the transferability of TriKD beyond MonoLift, we apply it to RT-1, which formulates policy learning as a multi-class classification task by discretizing continuous actions. Since RT-1 lacks continuous action distributions, we remove the action distribution distillation branch and retain only the spatial and temporal objectives. As shown in Table 5(b), applying only the spatial and temporal components of TriKD to RT-1 still yields performance gains, demonstrating the transferability of these objectives. However, the improvement is smaller than that observed in MonoLift, which benefits from the full combination of all three objectives: spatial, temporal, and action, that work together to enhance policy learning. In addition to validating the applicability of TriKD, these results also highlight the individual effectiveness of each distillation branch, as well as the complementary advantage of their integration.

**Impact of Depth Quality on Distillation Performance.** We investigate how the quality of estimated depth affects the effectiveness of our method. Specifically, we employ three variants from the Depth Anything V2 series, `vits`, `vitb`, and `vitl`, which offer progressively better depth estimation [14]. Additionally, we include real depth maps rendered by the simulator for comparison. Unless otherwise specified, `vitl` is used as the default estimator in all experiments. Student performance improves with teacher depth quality (Figure 5(c)), highlighting the role of accurate geometric cues. Notably, students trained with `vitl`-based supervision nearly match the performance of those trained with

real depth. We attribute the result to two main factors. First, `vitl` offers high-quality structural cues with clear boundaries and spatial continuity, enabling reliable guidance. Second, by distilling feature representations and action distributions rather than raw pixels, MonoLift is less affected by local errors and can leverage imperfect but structurally meaningful depth estimates.

**Deployment Efficiency Evaluation.** We evaluate deployment efficiency in terms of inference time and model size. As shown in Table 2, *MonoLift* incurs lower computational overhead than the *RGB+Depth* baseline. The latter adopts the same network architecture as *MonoLift* but takes both RGB images and estimated depth maps (from Depth Anything V2) as inputs, using the same fusion module as the teacher model to combine the two modalities. Unlike *MonoLift*, the *RGB+Depth* baseline requires depth inputs during both training and inference. This

Table 2: Deployment efficiency.

| Model | Time (ms) | Params (M) |
|---|---|---|
| RGB-only | 18.1 | 8.5 |
| RGB+Depth | 442.8 | 344.8 |
| **MonoLift** | 18.1 | 8.5 |

demonstrates the advantage of conducting depth-guided distillation without introducing additional runtime complexity. Compared to the *RGB-only* baseline, which is trained without depth or distillation, *MonoLift* delivers improved task performance without increasing deployment cost. Moreover, as reported in Table 4 in the appendix, our *MonoLift* consistently achieves higher success rates than the *RGB+Depth* baseline. These results confirm the efficiency and scalability of our method in resource-constrained environments.

## 5.3 Real-World Experiment

We conduct real-world experiments using a Franka Research 3 (FR3) arm, with data captured from a fixed frontal viewpoint using an Orbbec camera. The dataset covers six manipulation tasks: (1) press a button, (2) push a box into a goal, (3) pull out a tissue, (4) pick up grapes and place them in a fruit plate, (5) lift a cup and pour water, and (6) fold a towel. These tasks involve diverse interactive objects and manipulation types. For each task, we collect 10 demonstration trajectories covering varied spatial configurations, recorded at a frame rate of 20 FPS. To evaluate the model's robustness to natural variations and suboptimal executions, we intentionally do not clean the expert demonstrations, which include perception jitter and imperfect actions. The implementation details are consistent with our simulation experiments. During real-world evaluation, task success is determined through manual inspection, with each task evaluated over 10 trials.

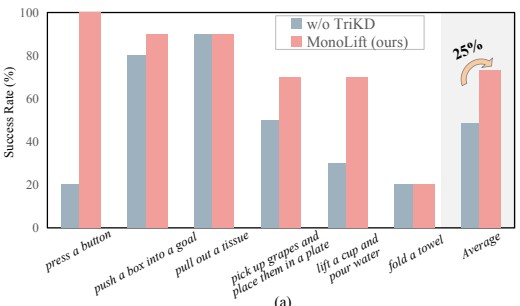 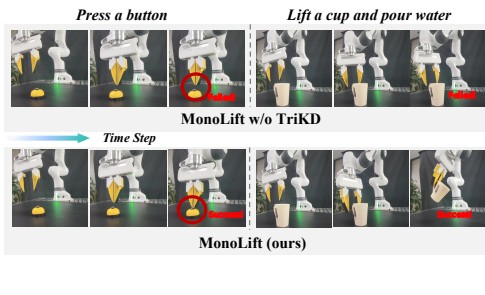

Figure 6: **(a) Quantitative comparison of real-world task success rates.** MonoLift achieves higher average performance than its ablated counterpart (w/o TriKD) across 6 real-world tasks. **(b) Qualitative visualization of real-world execution examples comparing MonoLift and w/o TriKD on 2 representative tasks.** (left) *press a button* and (right) *lift a cup and pour water*.

**Quantitative Results.** As shown in Figure 6 (a), we compare *MonoLift* with its ablated variant, *MonoLift w/o TriKD*, across six real-world manipulation tasks. Overall, MonoLift consistently achieves strong and reliable performance. For example, in the *lift a cup and pour water* task, which requires grasping a narrow handle and performing a smooth pouring motion, MonoLift demonstrates a clear performance advantage over its ablated variant. The *press a button* task involves pressing a small black area where even minor pose errors can lead to failure. In this high-precision task, MonoLift attains a perfect 100% success rate, while the baseline achieves only 20%. These improvements mainly result from our proposed TriKD framework, which transfers spatial, temporal, and action-level supervision from a depth-guided teacher. This design enables the student model to learn richer 3D

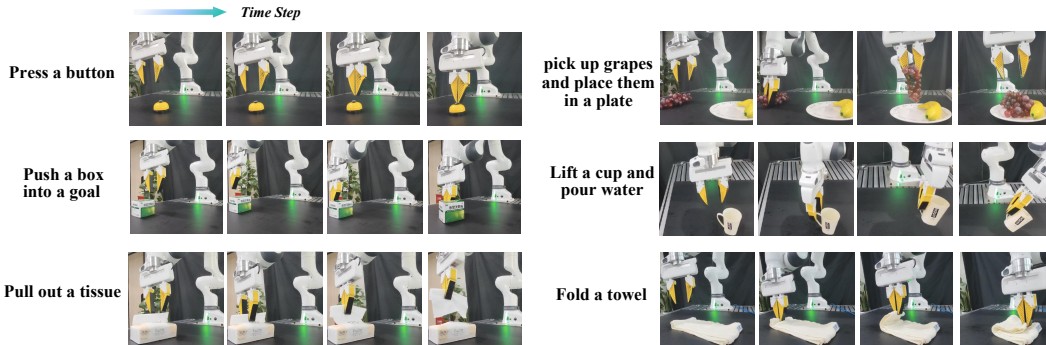

Figure 7: **Qualitative visualization of real-world task executions.** Trajectories from MonoLift are visualized for 6 tasks. Due to space constraints, only selected keyframes are shown rather than the full sequences.

representations and maintain stable control policies even under monocular RGB input. In contrast, the performance gap is smaller in the *fold a towel* task, where both methods achieve a 20% success rate. This is mainly because deformable object manipulation involves unstable geometry and weak depth cues, while the towel's uniform texture further weakens the guidance from the depth-guided teacher.

**Qualitative Results.** Figure 6(b) visualizes representative real-world executions. Consistent with the quantitative trends, MonoLift demonstrates stable and precise control behaviors across different tasks. In the *press a button* task, MonoLift precisely localizes the button and executes smooth end-effector motions, while w/o TriKD frequently misses or presses inaccurately. In the *lift a cup and pour water* task, MonoLift accurately aligns with the handle, performs a secure grasp, and smoothly modulates gripper rotation to achieve a controlled pour, showing fine-grained control over spatial alignment and motion timing. In contrast, w/o TriKD often misaligns with the handle due to its weaker 3D reasoning from monocular inputs. Figure 7 further presents successful executions across all six real-world tasks, illustrating the effectiveness and robustness of MonoLift in handling diverse manipulation tasks.

## 6    Conclusion

In this work, we propose MonoLift, an efficient framework for policy learning that bridges the gap between 2D perception and 3D action reasoning. Unlike prior approaches that rely on explicit 3D inputs or multi-stage pipelines to extract 3D cues, MonoLift adopts a training-only distillation strategy that leverages pseudo-depth maps generated by a pretrained depth estimator. This enables structured and efficient decision-making using only RGB inputs, without incurring any inference-time overhead. To facilitate 3D-aware policy learning, we adopt a tri-level knowledge distillation strategy that transfers spatial representations, temporal dynamics, and action distributions from a depth-guided teacher to a lightweight RGB-only student. Experimental results confirm that each distillation level contributes complementary benefits, leading to consistent improvements in overall performance. MonoLift provides a scalable and deployable solution for real-world robotic manipulation tasks, especially in scenarios where 3D sensors or heavy computational resources are unavailable.

**Limitations and Future Work.** Although MonoLift achieves strong performance across multiple tasks through its tri-level distillation strategy, it still exhibits a notable limitation. While the training phase incorporates a depth-guided teacher model, the framework does not explicitly model the uncertainty or potential errors in the estimated depth. In real-world settings, depth estimators could produce inaccurate or unstable outputs due to factors such as occlusion, specular reflection, transparent materials, or sensor interference. While MonoLift directly leverages estimated depth signals to guide the student's spatial and temporal representation learning, these signals may introduce slight noise or uncertainty, potentially posing challenges for geometric understanding under visually complex or ambiguous conditions. Future work could explore an uncertainty-aware distillation mechanism that adaptively adjusts the depth guidance based on its estimated reliability, enhancing the robustness in such challenging scenarios.

## Acknowledgements

This work was supported by the National Natural Science Foundation of China (Grant No.62403429) and the Natural Science Foundation of Zhejiang Province (Grant No.LQN25F030008).

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
