# OpenReview forum: "MonoLift: Learning 3D Manipulation Policies from Monocular RGB via Distillation"
_NeurIPS.cc/2025/Conference — NeurIPS 2025 spotlight_

### Official Review · Reviewer_fcUe · 2025-06-24

**Clarity:** 3
**Significance:** 2
**Originality:** 3
**Rating:** 4
**Confidence:** 5

**Summary:**

This paper presents MonoLift, a tri-level knowledge distillation framework that enables learning 3D manipulation policies from monocular RGB images by transferring spatial, temporal, and action-level knowledge from a depth-guided teacher to an RGB-only student model.

**Questions:**

Please refer to my comments above.

**Ethical Concerns:**

["NO or VERY MINOR ethics concerns only"]

**Final Justification:**

I appreciate the authors' diligent efforts in the rebuttal to address several concerns. However, key issues regarding the “module integration weakens innovation” remain inadequately resolved. Consequently, my rating stays at borderline accept.

**Limitations:**

Please refer to my comments above.

**Quality:**

3

**Strengths And Weaknesses:**

1.	While the paper claims to eliminate inference-time overhead, it still requires access to high-quality depth estimation during training. This dependency on pre-trained depth estimators may limit the method's applicability in scenarios where obtaining reliable depth estimates during training is challenging or expensive, somewhat undermining the core motivation of making 3D manipulation learning more accessible.
2.	The paper lacks theoretical analysis of what specific 3D knowledge is being transferred through the tri-level distillation and why this particular combination of spatial, temporal, and action distillation is optimal. While the empirical results are promising, the absence of theoretical justification makes it difficult to understand the fundamental principles and predict when the approach might fail.
3.	Although the results are consistent across tasks, the absolute performance gains are often modest. For instance, on Meta-World, MonoLift achieves 87.8% vs 83.2% for the best 3D baseline - a meaningful but not transformative improvement. The gains over RGB-only methods are more substantial but vary significantly across tasks, suggesting limited robustness.
4.	The demonstrated tasks, while spanning multiple benchmarks, are primarily focused on relatively standard manipulation scenarios. The evaluation lacks more complex manipulation tasks that would truly validate the claimed advantages of 3D-aware reasoning, and the real-world experiments are limited to basic manipulation primitives.
5.	While the tri-level distillation framework is well-designed, the individual components largely build upon existing techniques (standard depth estimation, transformer architectures, knowledge distillation). The primary contribution is the integration and application to manipulation learning rather than fundamental algorithmic innovations.

---

> ### Author Rebuttal · Authors · 2025-07-30
>
> Thank you for your detailed and constructive feedback. We appreciate your time and effort. Below, we respond to each of your comments and will incorporate the relevant revisions into the final paper.
> >While the paper claims to eliminate inference-time overhead, it still requires access to high-quality depth estimation during training. This dependency on pre-trained depth estimators may limit the method's applicability in scenarios where obtaining reliable depth estimates during training is challenging or expensive.
>
> Thanks for your thoughtful question. We clarify that our training setup is low-cost and robust to depth quality. We support this from the following perspectives:
>
> **Depth comes from reliable and off-the-shelf estimators.**
> Our method does not require depth ground truth during training. Instead, we obtain predicted depth from a publicly available monocular depth estimator trained on large-scale data, specifically Depth Anything v2 [1]. This large-scale trained estimator provides predicted depth without requiring specialized hardware, **generalizes robustly across datasets, and serves as a practical, inexpensive alternative**—as supported by recent work [2,3,4].
>
> **Our method remains effective with low-quality depth.**
> By distilling high-level spatiotemporal and behavioral patterns rather than pixel-level details, the student avoids overfitting to noisy regions—allowing imperfect depth to offer useful guidance. We confirmed this by testing with smaller, less accurate models; even with degraded depth quality, our method outperformed the RGB-only baseline, demonstrating robustness for real-world deployment.
> Depth Estimator|NYU-δ1↑| NYU-AbsRel↓|Success Rate
> -|-|-|-
> No depth|–|–|53.7
> S (low quality)|0.961|0.073|67.5(**+13.8%**)
> B (medium)|0.977|0.063|70.0(**+16.3%**)
> L (high quality)|0.984|0.056|80.8(**+27.1%**)
>
> *Note: δ1 and AbsRel are reported on NYU-D dataset following [1].*
>
> **Once-and-Done Depth Estimation.**
> In practice, we estimate depth once before training to prepare the teacher’s inputs. The depth estimator is not used afterward, keeping the training and inference pipeline lightweight.
>
> We will incorporate these clarifications into the revised version. Thank you again for pointing out this concern.
>
> [1] Yang L et al. Depth Anything v2. NeurIPS, 2024
>
> [2] Ren X et al. Gen3C. CVPR, 2025
>
> [3] Wang H et al. Levitor: 3D trajectory oriented image-to-video synthesis. CVPR, 2025
>
> [4] Izquierdo S et al. MVSAnywhere. CVPR, 2025
> >The paper lacks theoretical analysis of what 3D knowledge is transferred and why the chosen spatial, temporal, and action distillation combination is optimal. This limits understanding of the method’s principles and failure cases.
>
> Thank you for the thoughtful question. While a full theoretical analysis is challenging, we provide conceptual motivation and empirical evidence to support the rationale behind the chosen combination and help address your concerns.
>
> **Design Rationale and Supporting Evidence:**
> - Unlike static perception tasks, robotic manipulation involves dynamic decision-making shaped by spatial structure, temporal dynamics, and policy behavior—requiring accurate perception, temporal reasoning, and stable action generation. This motivates our tri-level distillation design across spatial, temporal, and action levels.
> - This formulation is also supported by prior work. For example, [1,2,3] shows that jointly modeling spatial and temporal features benefits dynamic tasks, and [4,5] demonstrate that multi-level distillation can significantly improve student model stability and generalization.
>
> **Empirical Validation:**
> - We validated each component through ablations (Fig. 5(a) in our paper), which show that removing any single objective leads to a drop in performance.
> - To further illustrate the role of each component, we added additional analytical experiments. While figures cannot be included in this rebuttal, we report three key metrics aligned with our distillation objectives: (1) Correlation between RGB features and depth (↑ 257%) — indicates how well the student captures 3D spatial structure., (2) Temporal difference (↑ 57.6%) — reflects the ability to capture dynamic information over time, and (3) Action std (↓ 19.1%) — measures policy stability.
>
> We believe that theoretical support also boosts the explanation, which is an interesting future direction but is not currently our focus.
>
> [1] Lin J et al. TSM. ICCV, 2019
>
> [2] Pan J et al. ST-Adapter. NeurIPS, 2022
>
> [3] Wang M et al. A multimodal, multi-task adapting framework for video action recognition. AAAI, 2024
>
> [4] Jin Y et al. Multi-level logit distillation. CVPR, 2023
>
> [5] Li M et al. Correlation-decoupled knowledge distillation for multimodal sentiment analysis with incomplete modalities. CVPR, 2024
>
> >Although the results are consistent across tasks, the absolute performance gains are often modest. For instance, on Meta-World, MonoLift achieves 87.8% vs 83.2% for the best 3D baseline-a meaningful but not transformative improvement. The gains over RGB-only methods are more substantial but vary significantly across tasks, suggesting limited robustness.
>
> We thank the reviewer for raising this point. **Performance gains vary depending on task difficulty, with greater improvements observed on harder tasks.**
>
> Using Meta-World as an example, we grouped tasks into three difficulty levels and observed the following patterns:
> * **Easy tasks** generally involve only coarse spatial reasoning and already perform well with RGB input, leaving little room for improvement.
> * **Medium tasks** show consistent and stable improvements with MonoLift.
> * **Hard tasks** benefit the most: they demand precise control and stronger spatial understanding. MonoLift outperforms the best 3D baseline by up to **+16.1%** in this category.
>
> *Notably, the strongest 3D baseline uses depth at inference, while MonoLift relies only on RGB—yet still outperforms it, especially on harder tasks, demonstrating both efficiency and effectiveness.*
>
> In addition, we evaluated the robustness of our method across different tasks by comparing the task-wise standard deviation of MonoLift with that of the best-performing 3D baseline:
> * **Meta-World**: MonoLift: **14.4%** vs. 3D Baseline: **25.7%**
> * **Libero-90**: MonoLift: **9.7%** vs. 3D Baseline: **12.9%**
>
> In both benchmarks, MonoLift **exhibits lower standard deviations across tasks**, demonstrating greater robustness.
> *We appreciate the opportunity to clarify this point and will incorporate this explanation in the final version of the paper.*
> >The tasks focus on standard manipulation scenarios, with limited evaluation on more complex ones that would better validate the benefits of 3D-aware reasoning. Real-world experiments also remain basic.
>
> We appreciate your comment and take this opportunity to clarify our benchmark selection and how task complexity is reflected in our evaluation.
>
> We chose Meta-World and Libero as benchmarks due to their **popularity and widespread use** in evaluating manipulation performance [1,2]. To introduce greater complexity, **we added Libero-Long, which requires long-horizon decision-making and fine-grained spatial reasoning**.
> MonoLift’s strong performance demonstrates its ability to handle complex tasks.
>
> Our real-world tasks **demand high spatiotemporal reasoning**, and we intentionally avoided cleaning or manual selection of expert demonstrations, **preserving noise** to better reflect real-world conditions. These challenges are reflected in the poor performance of the baseline, as shown in Appendix Figure 8. *Press a Button* is depth-sensitive—slight over-pressing can cause collisions. *Lift a Cup and Pour Water* demands accurate grasping and smooth pouring.
>
> This work focuses on developing the core method and validating it on widely adopted benchmarks. Informed by your feedback, we plan to extend it to more complex, unstructured settings in future work.
>
> [1] Yunhao Luo and Yilun Du. Grounding Video Models to Actions through Goal Conditioned Exploration. ICLR. 2025
>
> [2] Jia et al, Lift3D Foundation Policy. CVPR, 2025.
> >The tri-level distillation is well-designed but mainly integrates existing techniques, with the core contribution lying in application rather than novel algorithms.
>
> Thank you for the comment. We would like to clarify two core aspects of our method:
> 1. Our method is not a simple combination of existing components—it **extends prior work to address key challenges in dynamic manipulation**: weak spatial perception, limited temporal modeling, and unstable policies, which are respectively targeted by our tri-level distillation over spatial, temporal, and action dimensions.
>
>     Empirically, we found that **existing distillation strategies fail to address the core challenges**. We compared: (1) feature distillation, (2) action distillation, and (3) pretrain-and-freeze.
>     All variants underperformed due to key limitations: feature distillation weakens the perception–action link; action distillation misses intermediate features and cross-modal alignment; and pretrain+freeze prevents co-adaptation, limiting transfer.
>     -|No Distillation|Feature Distillation|Action Distillation| Pretrain+Freeze |MonoLift (Ours)
>     -|-|-|-|- |-
>     Libero-Long|46.5±1.3|61.0±1.6| 57.7±2.1|57.3±3.6|71.7±1.1
> 2. While monocular depth estimator can produce depth maps, **their direct use in manipulation policies is often inefficient and suboptimal**. We introduced an RGB+Depth baseline that uses both RGB images and depth maps (from Depth Anything v2) during both training and inference. Compared to MonoLift (8.5M / 18.1 ms), this baseline is significantly heavier (344.8M / 442.8 ms) and still underperforms—despite having richer inputs at inference time.
>
>     Method|Libero-90|Libero-Long|Meta-World
>     -|-|-|-
>     RGB + Depth|70.3±3.7%|53.0±2.4%|85.4±2.6%
>     MonoLift (ours)|80.8±3.3%|71.7±1.1%|87.8±2.3%

---

### Official Review · Reviewer_TXrx · 2025-06-30

**Clarity:** 3
**Significance:** 3
**Originality:** 3
**Rating:** 5
**Confidence:** 4

**Summary:**

This paper proposes MonoLift, a policy-distillation framework that applies tri-level knowledge distillation: (1) Spatial Distillation (matching the student’s RGB-only features to the teacher’s depth-enhanced features), (2) Dynamic Distillation (aligning temporal feature gradients), and (3) Action Distillation (minimising the KL divergence between student and teacher action distributions). The authors present an extensive experimental evaluation in both simulation and the real world, showing that MonoLift consistently outperforms a broad set of baselines.

**Questions:**

1. As mentioned in Weakness, what precisely is the “RGB + Depth” baseline in Table 2? Does it use the same architecture as MonoLift but with RGB + estimated depth inputs?
2. Using a small MLP as the policy head feels limiting, especially given the recent advances with ACT and Diffusion Policy heads. Implementing MonoLift with one of these stronger heads might further raise performance.
3. In the ablation study, did the authors remove the corresponding distillation loss in the Student for each of the ablated variations?
4. There is a typo `$M$ linear layers` in line 218.
5. A video showcasing the real-world experiments would help readers understand qualitative performance.
6. Classical distillation schemes first train (and freeze) the teacher before optimizing the student. Did the authors try this sequential alternative? What advantages led you to prefer the end-to-end joint approach reported in the paper?

**Ethical Concerns:**

["NO or VERY MINOR ethics concerns only"]

**Final Justification:**

The authors addressed most of my concerns in the rebuttal. I believe the paper is solid and has the potential to have a significant impact on the robot learning community, particularly in the field of policy learning using RGB-only inputs.

**Limitations:**

Although the authors discuss limitations, the section is relegated to the appendix. Moving it into the main text in the final version would improve transparency and balance.

**Quality:**

3

**Strengths And Weaknesses:**

Strength
1. The paper is well-motivated and timely: RGB-only perception is increasingly popular, and the proposed approach retains depth-related spatial cues during training while keeping test-time inference lightweight.
2. The experimental design is thorough, spanning several environments and a wide array of baselines. The empirical results are convincing.

Weakness

My main concern is the absence (or, at least, the lack of clarity) of a baseline that passes estimated depth to the policy at inference time instead of relying on distillation. The introduction claims that attaching a depth estimator `significantly increases computational cost and latency`, which is the core motivation for spatial distillation. Yet it is unclear whether such an RGB + Depth baseline is actually evaluated. Table 2 seems intended to quantify this cost, but it is not clear from lines 358–368 what this baseline contains. Moreover, including a clearly defined RGB + Depth baseline in the main quantitative results (Table 1 and Fig. 3a) would also let readers weigh accuracy against compute.

---

> ### Author Rebuttal · Authors · 2025-07-30
>
> Thank you for your thoughtful feedback—we appreciate your recognition of the motivation and the strength of our experimental design and results. We address each of your comments in detail below and will incorporate your suggestions into the final version of the paper.
> > Weakness: My main concern is the absence (or, at least, the lack of clarity) of a baseline that passes estimated depth to the policy at inference time instead of relying on distillation. The introduction claims that attaching a depth estimator significantly increases computational cost and latency, which is the core motivation for spatial distillation. Yet it is unclear whether such an RGB + Depth baseline is actually evaluated. Table 2 seems intended to quantify this cost, but it is not clear from lines 358–368 what this baseline contains. Moreover, including a clearly defined RGB + Depth baseline in the main quantitative results (Table 1 and Fig. 3a) would also let readers weigh accuracy against compute.
>
> This is a good point. We address this in two parts below.
>
> **Clarifying the RGB + Depth Baseline in Table 2:**
> The “RGB + Depth” baseline refers to a model that takes as input both RGB images and monocularly predicted depth maps (from Depth Anything v2) during both training and inference. It employs the same architecture as the teacher model used in MonoLift.
> Table 2 in the main paper reports its compute cost, including the inference-time depth prediction. Compared to MonoLift (8.5 M / 18.1 ms), the RGB+Depth baseline is much heavier (344.8 M / 442.8 ms), highlighting the substantial deployment overhead our method avoids.
>
> **Updated Results in the Main Table:**
> We originally reported this baseline in Table 2 to highlight compute cost. As suggested, we now include its performance in the main results table for a clearer comparison.
> We report the average success rates across benchmarks here:
> Method|Libero-90|Libero-Long| Meta-World
> -|-|-|-
> RGB + Depth|70.3±3.7%|53.0±2.4%|85.4±2.6%
> **MonoLift (ours)**|**80.8±3.3%**|**71.7±1.1%**|**87.8±2.3%**
>
> Although the RGB+Depth baseline benefits from richer inputs at inference time, it still underperforms MonoLift—especially on the most challenging Libero-Long benchmark. We attribute this to our multi-level distillation design, which makes more effective use of depth signals by transferring structured spatial, temporal, and behavioral knowledge into the RGB-only student.
>
> We will incorporate these comparisons and clarifications into the final version of the paper. Thank you again for this valuable suggestion—it significantly improves the completeness and clarity of our evaluation.
>
> >Q1:As mentioned in Weakness, what precisely is the “RGB + Depth” baseline in Table 2? Does it use the same architecture as MonoLift but with RGB + estimated depth inputs?
>
> Yes. The “RGB + Depth” baseline in Table 2 uses the **same architecture** as MonoLift, with both RGB and estimated depth maps (from Depth Anything V2) as inputs, along with the same simple fusion module used in our teacher model to combine the two modalities. Unlike MonoLift, this baseline requires depth inputs at both training and inference time.
>
> We will clarify this in the revised text to avoid confusion.
>
> >Q2:Using a small MLP as the policy head feels limiting, especially given the recent advances with ACT and Diffusion Policy heads. Implementing MonoLift with one of these stronger heads might further raise performance.
>
> Thank you for the insightful suggestion. Stronger policy heads such as those in ACT and Diffusion Policy are indeed effective. In our case, we adopted a lightweight MLP primarily due to efficiency and deployment constraints.
>
> | Policy Head      | Libero-Goal| Libero-Spatial| Libero-Object
> | - | -| -- | -
> | MonoLift (MLP)| 85.3±1.2%              | 78.6±0.9%     | 96.3±1.2%
> | MonoLift (Diffusion Policy)| 80.0±1.1% | 82.5±0.8%| 96.2±1.4%
>
> To test a more expressive head, we integrated a diffusion-style policy head into MonoLift. While not all tasks showed consistent improvement, **Libero-Spatial exhibited clear performance gains**, confirming that **MonoLift is compatible with more expressive policy heads and can benefit from them**.
>
> While the diffusion-style head improves task success on certain benchmarks, it also incurs noticeably higher inference latency—from **18.1ms** (MLP) to **221.6ms** (diffusion). That said, we recognize the rapid progress in accelerating diffusion inference [1,2,3], and see these advances as complementary to MonoLift. Exploring their integration is an exciting direction for future work.
>
> [1] Ma X et al. Deepcache: Accelerating diffusion models for free. CVPR, 2024: 15762–15772.
> [2] Huang X et al. Reverse transition kernel: A flexible framework to accelerate diffusion inference. NeurIPS, 2024, 37: 95515–95578.
> [3] Zhai Y et al. Motion consistency model: Accelerating video diffusion with disentangled motion-appearance distillation. NeurIPS, 2024, 37: 111000–111021.
>
> >Q3:In the ablation study, did the authors remove the corresponding distillation loss in the Student for each of the ablated variations?
>
> Yes, we removed the corresponding distillation loss for each ablated variant, confirming that each component individually contributes to performance.
>
> >Q4:There is a typo \$M$ linear layers in line 218.
>
> Thank you for your careful review and for pointing this out. We have corrected the formatting error in the revised version.
>
> >Q5:A video showcasing the real-world experiments would help readers understand qualitative performance.
>
> Thank you for the helpful suggestion. We will create a project homepage with videos of our real-world experiments and include the link in the final version of the paper to facilitate qualitative assessment.
> As external links are not allowed during the rebuttal phase, we will ensure the final version clearly references the video materials.
>
> >Q6:Classical distillation schemes first train (and freeze) the teacher before optimizing the student. Did the authors try this sequential alternative? What advantages led you to prefer the end-to-end joint approach reported in the paper?
>
> Thank you for the thoughtful question.
>
> We tested the standard sequential distillation setup: train and freeze the teacher, then train the student under fixed supervision. Despite using the same architectures, **joint distillation consistently outperforms sequential setup**. Additionally, sequential training significantly increases overall training time due to its two-phase structure, while our joint approach is simpler and more efficient to train.
>
> Setup|Libero-Long|Libero-Object
> -|-|-
> Sequential Distillation|57.3±3.6%|90.7±1.2%
> Joint Distillation|71.7±1.1%|96.3±1.2%
>
> Beyond empirical gains, joint training offers several benefits in our cross-modal distillation setting (RGB+Depth → RGB):
> 1. **Bridging the Modality Gap via Feature Alignment:**
>    In cross-modal distillation, the teacher and student operate on different input modalities, often leading to mismatches in feature distributions. Joint training helps bridge this modality gap by allowing both models to co-adapt their representations during optimization, improving alignment and transferability \[1,2].
> 2. **Adaptive, Student-Aware Supervision:**
>    Fixed supervision from a pretrained teacher fails to align with the student’s evolving capacity. Joint training enables the teacher to adapt its guidance based on the student’s learning progress, leading to more effective knowledge transfer—an effect also supported by prior studies [3].
>
> 3. **Improved Efficiency and Stability:**
>    Sequential distillation fixes the teacher, meaning it cannot adjust to the student’s training dynamics.  In contrast, the teacher and student are trained jointly, resulting in a more stable and efficient distillation process—particularly beneficial in noisy or long-horizon tasks [4].
>
> [1] Park D Y et al. Learning student-friendly teacher networks for knowledge distillation. NeurIPS, 2021, 34: 13292–13303.
> [2] Huo F et al. C2KD: Bridging the modality gap for cross-modal knowledge distillation. CVPR, 2024: 16006–16015.
> [3] Sengupta A et al. A good learner can teach better: Teacher-student collaborative knowledge distillation. ICLR, 2023: 25–29.
> [4] Ypsilantis N A et al. UDON: Universal dynamic online distillation for generic image representations. NeurIPS, 2024, 37: 86836–86859.
>
> > Limitations: Although the authors discuss limitations, the section is relegated to the appendix. Moving it into the main text in the final version would improve transparency and balance.
>
> Thank you for the suggestion. We agree and will move the limitations section from the appendix into the main text in the final version to improve transparency and completeness.

---

> > ### Comment · Reviewer_TXrx · 2025-08-01
> >
> > Thank you for the detailed rebuttal, most of my concerns are addressed.
> >
> > I am a bit surprised that MonoLift can achieve a better performance than RGB + Depth. Have you tried turning off other distillation and only keep the depth distillation? If so, how would this ablation compare with RGB + Depth?

---

> > > ### Author Response · Authors · 2025-08-01
> > >
> > > > Have you tried turning off other distillation and only keep the depth distillation? If so, how would this ablation compare with RGB + Depth?
> > >
> > > Thanks for your feedback. This is a good question. We indeed conducted experiments with the “only keep the depth distillation” variant, and the results are as follows:
> > >
> > > Method|Libero-90|Libero-Long
> > > -|-|-
> > > RGB-only (no distill)|53.7±3.5%|46.5±1.3%
> > > RGB + Depth|70.3±3.7%|53.0±2.4%
> > > only keep the depth distillation|69.6±3.2%|61.0±1.6%
> > > **MonoLift (ours)**|**80.8±3.3%**|**71.7±1.1%**
> > >
> > > On simpler short-horizon tasks (Libero‑90), directly using depth or distilling only depth spatial features both outperform the RGB-only baseline and achieve similar performance, while our full tri-level distillation performs best.
> > >
> > > On complex long-horizon tasks (Libero‑Long), the RGB+Depth improves over the RGB-only but is still outperformed by using only depth distillation, and the full tri-level distillation achieves the best results, driven by two key factors:
> > >
> > >   1. **Distillation mitigates depth noise accumulation.** RGB+Depth performs worse because using predicted depth in the inference introduces per-frame errors that accumulate over long sequences. In contrast, our distillation avoids direct reliance on predicted depth during inference and instead transfers high-level spatiotemporal and behavioral patterns rather than pixel-level details, mitigating the impact of depth noise.
> > >
> > >   2. **Temporal and action distillation bring additional benefits.**
> > >   (a) Temporal distillation transfers the teacher’s rich temporal cues, arising from its depth-informed perception that makes temporal changes more distinguishable and feature evolution more structured. This guidance helps the RGB-only student to detect subtle state transitions from visually similar frames and generate more reliable actions. This is consistent with prior findings that temporal modeling benefits dynamic tasks [1–3];
> > >   (b) Action distillation leverages the depth-informed teacher’s informative action distribution, preventing the RGB-only student from being distracted by irrelevant variations and producing unstable actions. This effect resembles the benefits of soft-label distillation reported in prior studies, which improve student performance and enhance stability by reducing output variance [4–6].
> > >
> > > [1] Lin J et al. Tsm: Temporal shift module for efficient video understanding. ICCV, 2019
> > >
> > > [2] Pan J et al. St-adapter: Parameter-efficient image-to-video transfer learning. NeurIPS, 2022
> > >
> > > [3] Wang M et al. A multimodal, multi-task adapting framework for video action recognition. AAAI, 2024
> > >
> > > [4] Phuong et al. Towards Understanding Knowledge Distillation. ICML, 2019.
> > >
> > > [5] Liang et al. Knowledge Consistency Between Neural Networks and Beyond. ICLR, 2019.
> > >
> > > [6] Zhou et al. Rethinking Soft Labels for Knowledge Distillation: A Bias–Variance Tradeoff Perspective. ICLR, 2021.

---

### Official Review · Reviewer_8Sa5 · 2025-07-01

**Clarity:** 3
**Significance:** 3
**Originality:** 3
**Rating:** 5
**Confidence:** 5

**Summary:**

In this paper, the authors propose a distillation framework for manipulation policy learning, where a teacher model that incorporates estimated depth, temporal dynamics is used; and a student model which only uses RGB image is used to distill the policy. The authors conduct extensive experiments on LIBERO-90, Meta-World and LIBERO-LONE. The proposed method outperforms baseline methods on these benchmarkes. Ablation studies are conducted to validate the model design choice.

**Questions:**

1. What are the weights of the three losses? How are they determined?How sensitive is the method to the weights?

2. More results of the teach models can be added, including performances, learning curves.

3. For the real-world experiments, the baseline performances are unexpected low. More explaination can be added.

**Ethical Concerns:**

["NO or VERY MINOR ethics concerns only"]

**Final Justification:**

My question have all been well addressed in the rebuttal. I'll keep my score of accept.

**Limitations:**

The limitations are discussed in the supp.

**Quality:**

3

**Strengths And Weaknesses:**

Stength:

The proposed method is simple yet effective, which can be promising and valuable for the community. The authors conduct extensive experiments and ablation studies to validate the effectiveness of the method.

The paper is well written and structurd. Important details are provided for reproducibility.

Weakness:
The effectiveness of the method on challenging tasks, where the depth estimation can be wrong, such as transparent and specular objects, is not shown.

---

> ### Author Rebuttal · Authors · 2025-07-30
>
> Thank you for your kind and positive feedback—we truly appreciate your recognition of our method’s effectiveness, clarity, and thorough experimentation.
> Below, we address each of your questions one by one, and we will carefully revise both the main text and the appendix according to your valuable suggestions.
> > Weakness: The effectiveness of the method on challenging tasks, where the depth estimation can be wrong, such as transparent and specular objects, is not shown.
>
> Thank you for this point. Transparent and specular surfaces are indeed challenging for robotic manipulation. While our method does not explicitly target such cases, its design helps reduce the impact of imperfect depth in three ways:
>
> * **Robustness via Depth-Free Inference**:
>   Our method uses a teacher with access to predicted depth during training, but the student relies only on RGB at test time. This avoids direct dependence on depth quality and improves robustness when depth predictions are unreliable.
>
> * **Imperfect Depth Still Provides Structure**:
>   Even when depth is noisy, it often preserves useful cues like occlusion boundaries or coarse geometry. The teacher can still extract spatial structure from this signal, allowing the student to benefit from it during distillation.
>
> * **Expert Supervision Remains a Core Signal**:
>   In addition to distillation, the student is trained with expert action labels via behavior cloning. This provides a strong and reliable learning signal, ensuring stable policy optimization.
>
> **Empirical results:** To evaluate robustness to varying depth quality, we used three pretrained depth estimators from Depth Anything v2 [1], corresponding to low-, medium-, and high-quality outputs. Even with the weakest model (S), the distilled RGB-only student clearly outperforms the RGB-only baseline. Performance further improves as depth quality increases, showing that our method is tolerant to lower-quality depth.
>
> | Depth Estimator      | NYU-δ1 ↑ | NYU-AbsRel ↓ |Success Rate|
> |- | - | - | - |
> | No depth (RGB-only baseline) | –        | –            | 53.7                     |
> | S (low quality)          | 0.961    | 0.073        | 67.5 (**+13.8%**)         |
> | B (medium)               | 0.977    | 0.063        | 70.0 (**+16.3%**)         |
> | L (high quality)         | 0.984    | 0.056        | 80.8 (**+27.1%**)         |
>
> *Note: δ1 and AbsRel are reported on NYU-D dataset following [1].*
>
> While our current work does not target transparent or specular objects, we agree this is an important challenge and plan to address it in future work.
>
> [1] Yang L et al. Depth Anything v2. NeurIPS, 2024, 37: 21875–21911.
>
> >Q1:What are the weights of the three losses? How are they determined?How sensitive is the method to the weights?
>
> Thank you for the question. We set the loss weights as follows: Spatial Distillation (1.0), Temporal Distillation (1.0), and Action Distribution Distillation (0.1). These weights were chosen based on empirical observations from preliminary experiments: we explored several reasonable configurations and found that this setting performs consistently well across benchmarks. We use a lower weight for action distribution distillation as it directly affects the final policy output and is more sensitive to the modality gap between RGB and RGBD inputs.
>
> To assess robustness to weight choices, we conducted ablation experiments on the Meta-World benchmark by varying each loss weight in {0.1, 1.0, 2.0}, keeping the other two fixed. The ablation study shows that **performance stays consistently above the non-distilled baseline (75.6%)** and is **stable across a range of weight values**, indicating robustness.
>
> Weight Setting|0.1|1.0|2.0|
> -|-|-|-|
> Spatial Representation Distillation|+8.0%|+12.2%|+11.7%
> Temporal Dynamics Distillation|+8.7%|+12.2%|+11.4%
> Action Distribution Distillation|+12.2%|+10.1%|+9.0%
>
> We will include these settings and results in the final version of the paper.
>
> >Q2:More results of the teach models can be added, including performances, learning curves.
>
> Thank you for the suggestion. In response, we provide results on the teacher models, focusing on two aspects.
>
> We first present results showing how different teacher models—equipped with pretrained depth estimators of varying quality from Depth Anything v2 [1]—affect student performance. As teacher quality improves, the distilled RGB-only student also achieves higher success rates.
>
> | Teacher Configuration  | NYU-δ1 ↑ | NYU-AbsRel ↓ |Success Rate|
> |- | - | - | - |
> | No depth (RGB-only baseline) | –        | –            | 53.7                     |
> | With Depth Estimator (S)  | 0.961    | 0.073        | 67.5 (**+13.8%**)         |
> |With Depth Estimator (B)| 0.977    | 0.063        | 70.0 (**+16.3%**)|
> |With Depth Estimator (L) | 0.984    | 0.056        | 80.8 (**+27.1%**)|
>
> In addition, we report the results of teacher model trained and tested directly with RGB and depth as input, without distillation. Interestingly, our MonoLift consistently outperforms the teacher across all benchmarks. This is mainly because: 1) Our tri-level distillation effectively transfers structured spatial, temporal, and action-level knowledge from the depth-guided teacher. 2) The student relies only on RGB at inference, avoiding direct dependence on noisy depth and improving robustness.
>
> Method|Libero-90 (%)|Libero-Long (%)| Meta-World (%)
> -|-|-|-
> Teacher model|70.3±3.7%|53.0±2.4%|85.4±2.6%
> **MonoLift (ours)**|**80.8±3.3%**|**71.7±1.1%**|**87.8±2.3%**
>
> We'll include the teacher training curves in the final appendix to show convergence and stability. Due to rebuttal constraints, we can't add figures here. Thank you again for the helpful suggestion—it helped us improve the completeness of the paper.
>
> >Q3:For the real-world experiments, the baseline performances are unexpected low. More explaination can be added.
>
> Thank you for the helpful observation. The relatively low success rates of the baseline model in our real-world experiments are largely attributable to two main factors:
>
> 1. **Real-World Tasks Demand High-Precision Localization:**
> Real-world tasks require accurate precise spatial reasoning and fine-grained temporal control, which are difficult for RGB-only models. The baseline struggle with weak 3D understanding, while our method improves robustness by distilling from a depth-informed teacher into an RGB-only student.
> This difference is clear in tasks like Press a Button and Lift a Cup and Pour Water. The first task involves pressing a small black area, where minor pose errors can cause failure. The second requires grasping a narrow handle and pouring smoothly.
>
> 1. **Imperfect Demonstrations Expose Weaknesses in the Baseline:**
> We didn’t clean the expert demonstrations to better reflect real-world conditions, including perception jitter and imperfect actions. This contrasts with some prior work that extracts only key frames for training [1]. RGB-only methods are more vulnerable to such noise, whereas our method learns more stable behavior through multi-level guidance during training.
>
> [1] Jia et al, Lift3D Foundation Policy: Lifting 2D Large-Scale Pretrained Models for Robust 3D Robotic Manipulation. CVPR, 2025.
>
> **We will include this analysis and discussion in the final version to better contextualize the real-world performance gap and highlight the benefits of our design.**
>
> >Limitations: The limitations are discussed in the supp.
>
> Thank you for the suggestion. We will move the main content of the limitations section into the main text to improve clarity and transparency.

---

> > ### Comment · Reviewer_8Sa5 · 2025-08-05
> >
> > Thank the authors for the explaination and additional experiments. All my concerns have been well addressed. I'll keep my score of accept. Please add the revision in the final version.

---

> > > ### Author Response · Authors · 2025-08-06
> > >
> > > Thank you for your positive feedback. We are glad that our responses have resolved your concerns.
> > > We will incorporate the revised content into the final version of the paper.

---

### Official Review · Reviewer_zdQB · 2025-07-04

**Clarity:** 3
**Significance:** 2
**Originality:** 3
**Rating:** 4
**Confidence:** 3

**Summary:**

This paper proposes MonLift, which learns a single-view manipulation policy with 3D awareness by distilling information from a teacher model during training. The proposed method uses three kinds of knowledge distillation:  spatial representation distillation by aligning feature to a teacher model that encodes both RGB and estimated depth; temporal dynamics distillation by aligning the temporal student feature towards teachers by matching the temporal gradients; the  action distribution distillation that matches the student's action distribution to the teacher's. During deployment, only the student policy with single-view image observation is needed.

Experiments are conducted on Libero and Metaworld benchmarks. The proposed method is shown to outperform several prior methods that use either explicit 3d information at test time or first reconstruct 3d information at test time instead of distilling the 3d knowledge at training time.

**Questions:**

- In figure 1 of the paper it is mentioned that  Multi-View RGB requires preprocessing and alignment. What preprocessing and alignment are needed? In my understanding, most modern multi-view RGM imitation learning algorithms do not need any specialized ways of fusing the multiview RGB images -- simply concatenating them channel wise or concatenating the feature vector is enough, e.g., as done in Diffusion Policy, OpenVLA, Pi-0, etc.
- Regarding Temporal dynamics distillation – an ablation on using l2 loss between teacher and student features instead of gradient matching may help validate this choice.
- Can the authors add a discussion on why is Action Distribution Distillation is needed if we already have expert action labels and the standard imitation learning loss using the expert action labels? Why additionally imitate the teacher actions would help?

**Ethical Concerns:**

["NO or VERY MINOR ethics concerns only"]

**Final Justification:**

The authors have cleared my concerns during the rebuttal and thus I have raised my score from borderline reject to borderline accept.

**Limitations:**

Yes

**Quality:**

3

**Strengths And Weaknesses:**

Strength:
- This paper is well-motivated. 3D information is essential for manipulation and how to infer that purely from single view RGB image is important.
- The paper is generally well-written and easy to follow.
- The experiments are relatively comprehensive and show good results of the proposed method.

Weakness:
- I think the paper is missing important comparison to some well-established multi-view RGB imitation learning baselines, such as Diffusion Policy, OpenVLA, and Pi-0. These methods use multi-view RGB images as input and has demonstrated strong performance on many imitation learning benchmarks. E.g. Table 12 in OpenVLA shows that OpenVLA and Diffusion Policy can achieve strong results on the Libero Benchmarkings. It is unclear how the proposed method compares to these stronger baselines, which makes it hard to access the true merits of the proposed method.
- Some details and ablations are missing from the paper. See the questions section below.

---

> ### Author Rebuttal · Authors · 2025-07-30
>
> Thank you for your feedback and recognition of our motivation, clarity, and experimental contributions. We have carefully responded to each point raised and provided a detailed reply to every comment.
> >W1: I think the paper is missing important comparison to some well-established multi-view RGB imitation learning baselines, such as Diffusion Policy, OpenVLA, and Pi-0. These methods use multi-view RGB images as input and has demonstrated strong performance on many imitation learning benchmarks. E.g. Table 12 in OpenVLA shows that OpenVLA and Diffusion Policy can achieve strong results on the Libero Benchmarkings. It is unclear how the proposed method compares to these stronger baselines, which makes it hard to access the true merits of the proposed method.
>
> Thank you for pointing this out. To address the concern, we added comparisons to Diffusion Policy and OpenVLA on the four benchmarks from Table 12 of the OpenVLA paper: Libero-goal, spatial, object, and long. Libero-long was already part of our original evaluation; the other three were newly added for completeness. For consistency, we report Diffusion Policy and OpenVLA results directly from Table 12 of the OpenVLA paper. **MonoLift achieves the highest average performance across these benchmarks.** Public evaluation results of Pi-0 on the Libero suite are currently unavailable, and due to the limited rebuttal timeline, we were unable to reproduce them reliably for a comparison.
>
> Method|Libero-Goal|Libero-Spatial|Libero-Object|Libero-Long|Average
> -|-|-|-|-|-
> Diffusion Policy from scratch|68.3±1.2%|78.3±1.1%|92.5±0.7%|50.5±1.3%|72.4%
> OpenVLA fine-tuned|79.2±1.0%|84.7±0.9%|88.4±0.8%|53.7±1.3%|76.5%
> MonoLift from scratch (ours)|85.3±1.2%|78.6±0.9%|96.3±1.2%|71.7±1.1%|**83.0%**
>
> We will incorporate the full comparison and analysis into the final version of the paper.
>
> >W2: Some details and ablations are missing from the paper. See the questions section below.
>
> Thank you for the comment. We address all questions below and will include the details and ablations in the final paper and appendix.
>
> >Q1: In figure 1 of the paper it is mentioned that Multi-View RGB requires preprocessing and alignment. What preprocessing and alignment are needed? In my understanding, most modern multi-view RGM imitation learning algorithms do not need any specialized ways of fusing the multiview RGB images -- simply concatenating them channel wise or concatenating the feature vector is enough, e.g., as done in Diffusion Policy, OpenVLA, Pi-0, etc.
>
> Thank you for the feedback. Our original intention was to refer to techniques for projecting and fusing multi-view inputs, including spatial projection, cross-view attention, and self-supervised feature fusion. We recognize that the original wording may have been ambiguous and will revise it to *“projection and fusion”* in the final version.
>
> It's worth noting that concatenation is a simple form of fusion. In Figure 1, our intent was to emphasize approaches that explicitly model inter-view relationships.
>
> For example, in embodied intelligence:
> - RVT [1] and HiveFormer [2] apply cross-view attention to enhance contextual modeling; SPA [3] aggregates multi-view features into unified 3D feature volumes; MV-MWM [4] uses self-supervised view masking and video reconstruction for improved 3D world modeling.
>
> Similar trends appear in other domains:
> - 3DMV-VQA [5] builds voxel-grid feature volumes from multi-view inputs; UniFusion [6] jointly models spatial and temporal information to improve cross-view understanding.
>
> *These approaches suggest that projection and fusion could lead to richer representations and improved performance, especially in tasks requiring complex spatial reasoning.*
>
>
>
> [1] Goyal A et al. RVT: Robotic view transformer for 3D object manipulation. CoRL, 2023: 694–710.
> [2] Guhur P L et al. Instruction-driven history-aware policies for robotic manipulations. CoRL, 2023: 175–187.
> [3] Zhu H et al. SPA: 3D spatial-awareness enables effective embodied representation. ICLR, 2025.
> [4] Seo Y et al. Multi-view masked world models for visual robotic manipulation. ICML, 2023: 30613–30632.
> [5] Hong Y et al. 3D concept learning and reasoning from multi-view images. CVPR, 2023: 9202–9212.
> [6] Qin Z et al. UniFusion: Unified multi-view fusion transformer for spatial-temporal representation in bird’s-eye-view. ICCV, 2023: 8690–8699.
>
>
> >Q2: Regarding Temporal dynamics distillation – an ablation on using l2 loss between teacher and student features instead of gradient matching may help validate this choice.
>
> Thank you for the suggestion. We performed this ablation study during the model design phase and are glad to present the results here.
>
> Specifically, we replaced the temporal gradient matching term in MonoLift with a standard L2 loss and evaluated the model on Libero-Long and Libero-Object. As shown below, **temporal gradient loss consistently outperforms L2 loss across both benchmarks**.
>
> -|Libero-Long|Libero-Object
> -|-|-
> L2 Loss|62.3±1.2%|93.7±0.9%
> Temporal Gradient Loss (ours)|**71.7±1.1%**|**96.3±1.2%**
>
> These results highlight a key insight: **static feature alignment (e.g., via L2 loss) is insufficient for capturing temporal dynamics critical to manipulation tasks**. With RGB-only input, adjacent frames often appear visually similar, making it difficult to detect meaningful state changes—this can lead to unreliable actions. In contrast, our temporal gradient loss proves more effective, enabling the student model to track how features evolve over time, thereby improving performance.
>
> We will include these findings in the final version  to clarify the benefit of our design.
>
> >Q3:Can the authors add a discussion on why is Action Distribution Distillation is needed if we already have expert action labels and the standard imitation learning loss using the expert action labels? Why additionally imitate the teacher actions would help?
>
> While standard imitation learning uses expert action labels as direct supervision, we introduce Action Distribution Distillation (ADD) to enhance policy learning in several ways:
>
> **Learning from a More Informed Teacher:** While expert labels offer deterministic targets, they convey limited information regarding uncertainty and the interdependencies among possible actions. In contrast, a teacher model equipped with RGB-D input yields a more informative action distribution, shaped by a deeper spatial understanding.
> Distilling this distribution helps the student acquire both action preferences and uncertainty estimates, similar to how soft labels have been proven effective in enhancing student models in previous studies [1,2].
>
> **Enhancing Policy Stability:**
> Single-view RGB policies are often sensitive to irrelevant appearance changes due to the lack of geometric cues, resulting in unstable actions. Our RGB+Depth teacher incorporates richer 3D structure to produce more robust and consistent actions, guiding the student via ADD as a form of regularization.
> To support this, we measured action variance under fixed inputs: the teacher exhibited a lower standard deviation (0.17) compared to the student (0.21), indicating smoother responses. This effect is supported by prior findings that soft labels in distillation reduce output variance [3].
>
> **Empirical Support:**
> Ablation results in Figure 5(a) in the main text confirm the effectiveness of ADD—removing it leads to performance drops, highlighting its additional benefit beyond standard imitation learning loss.
>
> [1] Phuong et al. Towards Understanding Knowledge Distillation. ICML, 2019.
>
> [2] Liang et al. Knowledge Consistency Between Neural Networks and Beyond. ICLR, 2019.
>
> [3] Zhou et al. Rethinking Soft Labels for Knowledge Distillation: A Bias–Variance Tradeoff Perspective. ICLR, 2021.

---

> > ### Comment · Reviewer_zdQB · 2025-08-03
> > **Response**
> >
> > Thank you for the detailed response to my questions. My concerns are mostly addressed. I have raised my score accordingly.

---

> > > ### Author Response · Authors · 2025-08-03
> > >
> > > Thanks for your positive feedback and the updated score. We truly appreciate it.

---

### Note · Authors · 2025-08-11

We sincerely thank all reviewers for their time and constructive feedback on our manuscript. We greatly appreciate that the reviewers recognized the importance and value of our work, highlighting strengths such as:

- the **well-motivated nature, value, and timeliness** of our work — centered on enabling 3D-aware manipulation purely from monocular RGB inputs (Reviewers zdQB, 8Sa5, TXrx);

- the proposed method being **streamlined, effective, and efficient**, well-organized, clearly written, and **providing considerable contributions to the field** (Reviewers zdQB, 8Sa5, TXrx);

- the **thorough experimental evaluation, with convincing results** validating our design (Reviewers zdQB, 8Sa5, TXrx).

We also appreciate that **the reviewers who actively engaged with us during the rebuttal phase acknowledged that their initial concerns were addressed in detail**. We kindly hope that the AC will take into consideration the overall positive feedback together with these discussions when making the final decision.

---

### Decision · Program_Chairs · 2025-09-17

**Decision:**

Accept (spotlight)

**Comment:**

This paper proposes MonoLift, a tri-level distillation framework that transfers spatial, temporal, and action-level knowledge from a depth-guided teacher to a monocular RGB student, enabling efficient 3D-aware manipulation without explicit depth at deployment. The method is well motivated and clearly written, and the experiments across LIBERO-90, LIBERO-LONE, Meta-World, and real-world tasks are comprehensive. Results show MonoLift consistently outperforms monocular baselines and even some methods using explicit 3D input.

**Strengths**:
- Clear writing and strong motivation (Reviewers zdQB, TXrx).
- Comprehensive experiments, with ablations validating design choices (Reviewers zdQB, 8Sa5, TXrx).

**Weaknesses**:
Real-world results are less convincing compared to simulation ones, with no videos provided.

**Rebuttal**:
The authors provided a thorough response with additional experiments, including:
- Comparisons with multi-view baselines (Diffusion Policy, OpenVLA).
- Ablations on temporal dynamics distillation, teacher depth quality, distillation strategies, and head architectures.
- An RGB+depth baseline.

These additions are insightful and address reviewer concerns, though real-world validation could still be strengthened.

**Suggestions for Improvement**:
- Include videos showcasing real-world experiments (Reviewer TXrx).
- Incorporate additional experiments into the main paper or appendix for clarity and completeness.